# Guideline Learning for In-Context Information Extraction

**Chaoxu Pang**[1,2]**, Yixuan Cao**[1,2*]**, Qiang Ding**[1,2]**, Ping Luo**[1,2,3*]

[1]Key Lab of Intelligent Information Processing of Chinese Academy of Sciences (CAS)
Institute of Computing Technology, CAS, Beijing 100190, China
[2]University of Chinese Academy of Sciences, Beijing 100049, China
[3]Peng Cheng Laboratory, Shenzhen 518066, China
{pangchaoxu21b,caoyixuan,dingqiang22z,luop}@ict.ac.cn

## Abstract

Large language models (LLMs) can perform a new task by merely conditioning on task instructions and a few input-output examples, without optimizing any parameters. This is called In-Context Learning (ICL). In-context Information Extraction (IE) has recently garnered attention in the research community. However, the performance of In-context IE generally lags behind the state-of-the-art supervised expert models. We highlight a key reason for this shortfall: *underspecified task description*. The limited-length context struggles to thoroughly express the intricate instructions and various edge cases of IE tasks, leading to misalignment in task comprehension with humans. In this paper, we propose a *Guideline Learning* (GL) framework for In-context IE which reflectively learns and follows guidelines. During the learning phrase, GL automatically synthesizes a set of guidelines based on a few error cases, and during inference, GL retrieves helpful guidelines for better ICL. Moreover, we propose a self-consistency-based active learning method to enhance the efficiency of GL. Experiments on event extraction and relation extraction show that GL can significantly improve the performance of in-context IE.

## 1 Introduction

Information extraction (IE), whose primary goal is to extract structured information from unstructured plain text, serves as a critical foundation for numerous downstream tasks such as question answering and knowledge base construction ([Wang et al., 2022a](#); [Fei et al., 2022](#)). IE tasks typically have complex task settings due to their requirement of translating diverse real-world facts into a few predefined classes. This often necessitates a large number of rules and examples to thoroughly and accurately define the *target concept* of the task. For example, the guidelines for ACE relation extraction

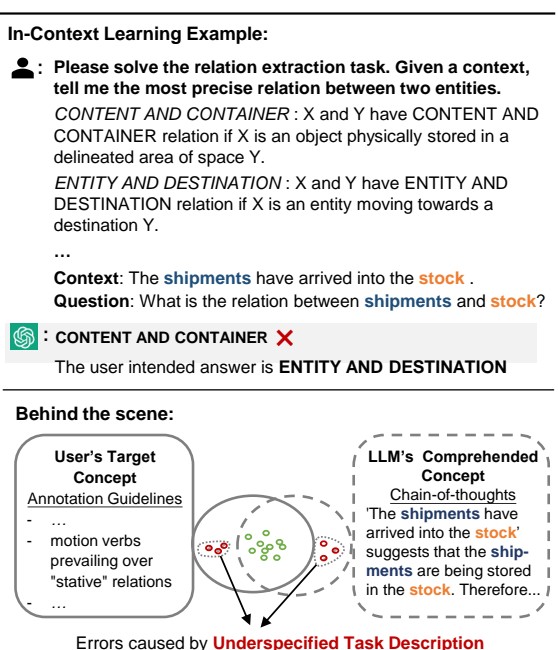

Figure 1: An example of *conceptual bias* in the relation classification task (SemEval 2010 Task 8).

extend over 33 pages ([Consortium, 2008](#)). In the past, the supervised learning paradigm has been applied to fine-tune numerous parameters on massive data to accurately learn the concept ([Li et al., 2020](#); [Zheng et al., 2019](#)). This approach, while effective, is data-intensive, hard to train, and difficult to update.

Recently, however, the NLP community witnesses the rapid rise of large language models (LLMs), such as PaLM ([Chowdhery et al., 2022](#)), ChatGPT ([OpenAI, 2023a](#)) and LLaMA ([Touvron et al., 2023](#)). These LLMs have achieved great performance on a wide range of NLP tasks with their superior language understanding power, but fine-tuning them faces closed-source and high-training-cost issues. In-Context Learning (ICL) ([Brown et al., 2020](#)), a characteristic feature of LLMs, offers a solution to harness the power of LLMs while

---

*Corresponding author: Yixuan Cao and Ping Luo.

sidestepping these issues. ICL enables LLMs to perform new tasks without tuning any parameters. Instead, they are given only the task instruction and a few input-output examples as the prompt. It achieves promising performance on many tasks like natural language inference and sentiment classification (Brown et al., 2020), demonstrating a new paradigm in the NLP community.

Several recent studies have explored the ICL paradigm for IE (Han et al., 2023; Wei et al., 2023). Impressively, by merely providing task instructions and a handful of in-context examples, LLMs can achieve significant performance on many IE tasks. However, they still lag behind supervised SOTA models (Han et al., 2023).

We underline one primary reason for the suboptimal performance: *underspecified task description*. As discussed earlier, the *target concept* of IE is inherently complex. But the input context utilized for elucidating the target concept to LLMs is constrained by its limited length. Consequently, the *comprehended concept* by LLMs might deviate from the target concept. An example of this is illustrated in Figure 1. In the sentence "The shipments have arrived into the stock", the pre-defined relation types Content-Container and Entity-Destination presents a grey area concerning the relation between the entities "shipments" and "stock". The target concept is embodied in a rule in the annotation guidelines[1] - "motion verbs prevailing over stative relations" - which is misaligned with the LLM's comprehended concept.

This paper attempts to mitigate this problem by introducing a *Guideline Learning* (GL) framework. This framework replicates the human annotation process, which first gathers annotation guidelines, and then annotates accordingly. Specifically, it has two phrases. In the learning phase, a set of *guidelines* are iteratively learned from scratch based on a few labeled instances. A guideline here is a natural language rule derived by integrating the appropriate extrapolation of an error instance and its true label. This is different from previous supervised learning methods, which learn a set of model parameters. In the inference phase, given a new instance, it retrieves relevant rules from the guideline to compose a prompt, which includes the task instruction, the retrieved rules, a few examples, and the input instance. It then asks an LLM agent to finish the task given the prompt. This failure-driven remind-

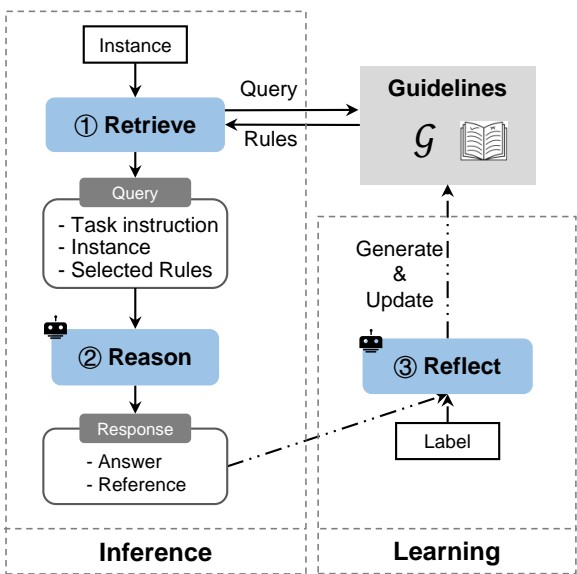

Figure 2: The Guideline Learning Framework, including inference and training phrases. 🤖 denotes LLM agents are applied in this phrase.

ing mechanism, similar to Madaan et al. (2022), is inspired from the theory of recursive reminding in psychology (Jacoby and Wahlheim, 2013). This theory suggests that human learn from the error cases and recall the most helpful experiences when encountering a new case.

Furthermore, we incorporate a self-consistency-based active learning method to enhance the efficiency of label utilization. we also propose a "generalizer" to assist in the generation and retrieval of guidelines. Finally, we conduct in-depth experiments on two representative IE tasks: (1) event extraction on financial documents, and (2) relation extraction on general domain resources, which both feature relatively complex target concepts. Experimental results indicate that the use of 50 labeled samples per class can greatly boost the performance of ICL in both tasks.

## 2 Guideline Learning Framework

### 2.1 Overview

Figure 2 presents an overview of the Guideline Learning (GL) framework. For the **inference phase**, assuming we have collected a set of guidelines for a task. Given an input instance $x$, the GL framework first retrieves a set of relevant rules from the guidelines. A query is constructed by assembling the task instruction, few in-context examples, the instance, and the retrieved rules. The query is then forwarded to an LLM agent, which generates

---

[1]Data Creation Guidelines for the SemEval 2010 Task 8

both the answer and the references (rules that the agent deems beneficial for this particular instance). During the **training phrase**, the framework iterates over a few training instances to generate and learn guidelines from scratch. For each instance, if the predicted answer from the LLM agent is different from the annotation, another LLM agent generates a new rule and update the existing guidelines.

In the following sections, we will detail the inference phrase (Sec 2.2), the learning algorithm (Sec 2.3), and an active instance selection method for effective guideline learning (Sec 2.4).

## 2.2 Inference

In this section, we introduce how to predict the answer of an instance $x$ in the GL framework. Suppose we have collected the **Guidelines** $\mathcal{G} = \{r_i\}|_{i=1}^{|\mathcal{G}|}$ which is a set of rules that supports read, write, and retrieve operations. Each rule, expressed as a natural language description, explicates an aspect of the task, while the guidelines implicitly reflect the *target concept* of the task. The inference process unfolds as follows.

**Retrieve**. We retrieve the top-k rules $R$ from $\mathcal{G}$ that are most relevant to $x$:

$$R = \text{Retrieve}(x, \mathcal{G})$$

where $R \subset \mathcal{G}$. We can also retrieve some input-output examples $N$ from the training dataset $\mathcal{D}$.

**Reason**. The task instruction $\mathcal{T}$, the instance $x$, the few-shot examples $N$, and the retrieved rules $R$ are integrated to create a query $q$, which is used to ask the reasoner about which class the instance belongs to:

$$q = f(\mathcal{T}, x, R, N), \quad \hat{y}, R^* = \text{Reason}(q)$$

where reasoning is performed by an LLM agent with ICL capability, $\hat{y}$ is the predicted answer, and $R^* \subset R$ is a returned subset of retrieved rules that the agent deems helpful during reasoning. $R^*$ is used to evaluate the quality of the rules in Sec 2.3.

## 2.3 Learning Algorithm

In this section, we introduce the learning algorithm which reflectively learns guidelines from a collection of instance-label pairs. The pseudo code of this algorithm is presented in Algorithm 1. In each epoch, we first predict on all instances to get the response comprising the answer $\hat{y}$ and references $R^*$. If the answer is wrong, an LLM agent will generate a new guideline and append it in a cache. We don't

update guidelines immediately to ensure stable reasoning inside one epoch. After the iteration, we update rules in the cache to the guidelines. Besides, we keep a score for each rule based on whether it leads to correct answers. At the end of an epoch, rules with a score below a threshold are regarded as harmful and are removed from the guidelines.

Specifically, the rules are generated as follows. If the predicted answer $\hat{y}$ is wrong, the instance $x$, the predicted $\hat{y}$, and the true label $y$ are given to an LLM agent to write a rule:

$$r = \text{Reflect}(x, \hat{y}, y)$$

The score of a rule is computed as follows. For a rule $r \in \mathcal{G}$, we compute its prior score based on its statistics:

$$\text{score}(r) = \frac{N_{hit} - N_{wrong}}{N_{retrieve}}$$

where $N_{retr}$, $N_{hit}$, and $N_{wrong}$ are the number of instances in which the model retrieves $r$ ($r \in R$), refers to $r$ ($r \in R^*$) and predicts correctly, and refers to $r$ and predicts wrongly. The prior score indicates the helpfulness of a rule based on the historical responses.

---

**Algorithm 1:** Guideline Learning

**Input** : number of epoch $N_e$, task description $\mathcal{T}$, training set $\mathcal{D} = \{(x_m, y_m)\}_{m=1}^{N_d}$

**Output** : guidelines $\mathcal{G}$

1 Initialize $\mathcal{G} = \emptyset$, cache $= \emptyset$;
2 **for** $e = 1...N_e$ **do**
3     **for** $(x, y)$ *in* $\mathcal{D}$ **do**
4         $R = \text{retrieve}(x, \mathcal{G})$;
5         $N = \text{retrieve\_examples}(x, \mathcal{D})$;
6         $q = f(\mathcal{T}, x, R, N)$;
7         $\hat{y}, R^* = \text{reason}(q)$;
8         $\text{update\_score}(\mathcal{R}^*, \hat{y}, y, \mathcal{G})$;
9         **if** $\hat{y} \neq y$ **then**
10             $r = \text{reflect}(x, \hat{y}, y)$;
11             cache $=$ cache $\cup \{r\}$;
12     **foreach** $r \in cache$ **do**
        $\text{update\_guideline}(r, \mathcal{G})$ ;
13     $\text{forget\_harmful\_guidelines}(\mathcal{G})$;

---

## 2.4 Active Instance Selection

In this section, we investigate how to select instances for annotation, to construct the training

dataset for effective guideline learning (Sec 2.3). Random sampling could result in a low efficiency as the model may already be capable of accurately predicting a large portion of instances. To alleviate this problem, we propose an active learning approach that prioritizes instances where the model is most uncertain.

Assume we have a collection of instances $\mathcal{I} = \{x_m\}_{m=1}^{|\mathcal{I}|}$. Following self-consistency chain-of-thoughts (Wang et al., 2022b), for each instance $x$, we first sample $T$ reasoning paths and answers $\{(r_t, \hat{y}_t)\}_{t=1}^T$ with a relatively high temperature. Then we obtain the model's probability on each class $c$ by marginalizing out the sampled reasoning paths:

$$p(c|x) = \frac{1}{T} \sum_{t=1}^{T} \mathbb{I}\{\hat{y}_t = c\}$$

The consistency of the sampled answers indicates the model's confidence. A sharp probability distribution indicates a high confidence on this instance, whereas a flatten distribution indicates a low confidence. We compute the negative entropy of the probability distribution to measure the model's confidence on this instance:

$$\text{confid}(x) = \sum_c p(c|x) \log p(c|x)$$

We select the top-k instances with the lowest confidence score. The underlying assumption here is that the model is more prone to committing errors for instances with lower confidence.

## 3 Task and Implementation

Initially, we implement the guideline learning framework for two information extraction tasks: event extraction (Sec 3.1) and relation extraction (Sec 3.2). We choose these tasks because the *target concepts* of these tasks are relatively complex.

### 3.1 Event Extraction

Event extraction (EE) aims to extract structured events from unstructured texts. Figure 3 gives an example of EE. The event structure is predefined by an event schema, consisting of event classes and corresponding event roles. For example, the *equity repurchase* event has roles like *company name*, *repurchased shares*, *closing date*, etc. In this paper, we decompose EE into three sequential sub-tasks:

1. **event trigger identification** (ETI) that identifies all candidate event triggers from the text;

2. **event trigger classification** (ETC) that classifies candidate event triggers to event classes;

3. **event argument extraction** (EAE) that identifies the event arguments of a given trigger and recognize the specific roles they play.

For this task, we apply guideline learning to ETC. Specifically, given an event schema and a set of candidate triggers in a text, one **instance** here is the text and one candidate trigger. Note that it's also feasible to apply guideline learning to EAE. We leave it as future work.

### 3.2 Relation Extraction

Relation extraction (RE) aims to predict semantic relations between a pair of entities in texts. Figure 1 presents an example of RE. According to a recent report (Han et al., 2023), even when equipped with chain-of-thought prompting, ChatGPT can only achieve a maximum performance of 43% compared to state-of-the-art RE methods.

For RE, we directly apply guideline learning to assist in distinguishing relation concepts. Specifically, given a set of relation types and one entity pair from a text, one **instance** here is the text and one entity pair.

### 3.3 Implementation of Base Components

**LLM Agent** ♔ For all LLM agents, we use the official API[2] of ChatGPT (OpenAI, 2023a) to generate outputs. To prevent the influence of dialogue history, we generate the response separately for each testing sample.

**Generalizer** We introduce an important LLM agent *generalizer* to narrow the shallow semantic gap between instances and rules. The generalizer is an LLM agent which extrapolates the instance $x$ properly to a more general form $\tilde{x}$ by abstracting common properties, such as company names, dates. We use the $\tilde{x}$ instead of $x$ to retrieve and generate rules. Figure 3 presents an example of the generalizer in EE. We provide some intuition of the generalizer in Appendix A.3.

**Retrieval** For an input instance, we use its general form $\tilde{x}$ to sort rules in guidelines by the semantic similarity between $\tilde{x}$ and the rules. Specifically, we use the embedding API (text-embedding-ada-002) from OpenAI (2023b) to obtain the embeddings of $\tilde{x}$ and $r$, and use cosine similarity as the

---

[2]gpt-3.5-turbo-0301.

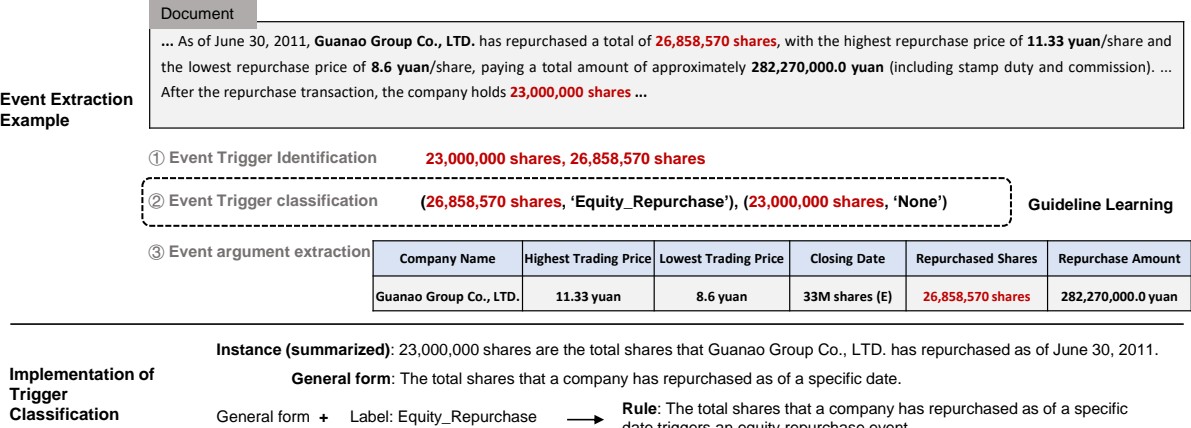

Figure 3: An example (translated) of event extraction from ChFinAnn dataset (Zheng et al., 2019). We decompose EE into three sub-tasks: event trigger identification, event trigger classification, event argument extraction. We present the output of each sub-tasks.

semantic similarity score. The few-shot demonstrations are randomly chosen from the training data, and are fixed for all instances and methods in each task.

**Reflect**  In this paper, we simply concatenate the general form $\tilde{x}$ of the instance $i$ and the golden label to generate a rule. Figure 3 presents an example of this process in EE.

Note that our implementation only requires the official APIs without any parameter updating.

## 4  Experiments

We conduct experiments[3] to demonstrate the effectiveness of the GL framework on event extraction (Sec 4.1) and relation extraction (Sec 4.2). In the last section, we analyze the quality of learned guidelines and conduct case studies (Sec 4.3).

### 4.1  Event Extraction

#### 4.1.1  Setup

**Dataset**  We use the ChFinAnn dataset (Zheng et al., 2019), a distant-supervised document-level event extraction dataset on Chinese financial documents, to conduct our experiments. Zheng et al. (2019) highlighted one challenge is to detect multiple event instances in one document. We focus on four event types: *Equity Freeze* (EF), *Equity Repurchase* (ER), *Equity Underweight* (EU), and *Equity Overweight* (EO). For the test set, We randomly sample at most 200 documents with proper token length for each event type from the original test

set due to the token length limit of OpenAI's API. More details are presented in Appendix A.1.1.

**Metrics**  We use role-level micro precision, recall, and F1 for evaluation, following previous work (Zheng et al., 2019).

**Method**  Though only working on ETC, we also provide simple solutions for the other two subtasks for comparison with other methods. Specifically, for ETI, as all event types are related to equity transaction, we identify text spans with the format "{number} shares" as candidate triggers via string matching. For ETC, we apply guideline learning framework and conduct binary classifications for each event type. As the documents in this dataset are long, we apply an extra LLM agent to generate a description for each trigger about its meaning according to the document. We use the generated description as the input instance to conduct the classification. For EAE, we apply an LLM agent to generate an event table in the markdown format given predicted event triggers.

**Compared Models**  (1) **ReDEE** (Liang et al., 2022) and **DE-PPN** (Yang et al., 2021): Two supervised methods. We reproduce DE-PPN on the entire dataset strictly following the official code. ReDEE runs out of memory on 12G GPU so we do not reproduce it. (2) **EE-ICL**: Prompt the LLM to directly output the event table without predicting event triggers. (3) **EE-GL-b**: Baseline version of our guideline learning method with empty guidelines. (4) **EE-GL-r**: Our guideline learning method. We *randomly* sample 50 documents from the training set and annotate event triggers. (5) **EE-GL-a**: We *actively* select 50 documents out of 400

---

[3]All prompts and hyper-paramter settings are detailed in the Appendix. All datasets and our annotations are publicly available for research purposes here.

| Method | EU | | | ER | | | EO | | | EF | | |
|---|---|---|---|---|---|---|---|---|---|---|---|---|
| | P. | R. | F1. | P. | R. | F1. | P. | R. | F1. | P. | R. | F1. |
| **DE-PPN†** | 69.7 | 79.9 | 74.4 | 91.1 | 89.3 | 85.6 | 87.4 | 81.0 | 71.3 | 78.2 | 69.4 | 73.5 |
| **ReDEE†♣** | 82.5 | 69.2 | 75.3 | 91.1 | 90.3 | 90.7 | 83.7 | 73.1 | 78.1 | 78.0 | 70.6 | 74.1 |
| **DE-PPN♠** | 71.2 | 66.1 | 68.6 | 84.3 | 88.2 | 86.2 | 70.9 | 71.9 | 71.4 | 72.6 | 56.0 | 63.2 |
| **EE-ICL** | 51.8 | 74.0 | 60.9 | 85.2 | 88.4 | 86.8 | 60.4 | 75.9 | 67.3 | 43.2 | 65.6 | 52.1 |
| **EE-GL-b** | 54.3 | 71.0 | 61.5 | 85.0 | 89.3 | 87.1 | 62.0 | 74.6 | 67.7 | 44.7 | 63.5 | 52.5 |
| **EE-GL-r** | **56.3** | 72.6 | 63.4 | 86.5 | **89.4** | 87.9 | **66.5** | 74.0 | 70.1 | 45.2 | **66.7** | 53.9 |
| **EE-GL-a** | 55.0 | **76.0** | **63.8** | **86.7** | 89.2 | **88.0** | 65.8 | **76.2** | **70.6** | 48.6 | 66.6 | **56.2** |

Table 1: Overall event-level precision (P.), recall (R.) and F1 scores evaluated on the test set (distant-supervised label). †: results from Liang et al. (2022). Note that these performances are not comparable as they evaluate on the entire test set. ♣: SOTA supervised model. ♠: We reproduce this work following Yang et al. (2021).

| Method | Single | | | Multi | | | All | | |
|---|---|---|---|---|---|---|---|---|---|
| | P. | R. | F1. | P. | R. | F1. | P. | R. | F1. |
| **DE-PPN** | 78.7 | 79.8 | 79.3 | 72.9 | 42.2 | 53.4 | 73.9 | 57.1 | 64.4 |
| **EE-ICL** | 64.6 | 88.9 | 74.8 | 70.8 | 79.4 | 75.0 | 68.1 | 83.3 | 74.9 |
| **EE-GL-b** | 71.5 | 87.8 | 78.8 | 73.0 | 72.2 | 72.6 | 72.8 | 77.9 | 75.3 |
| **EE-GL-r** | **72.4** | 88.7 | **79.7** | **74.4** | 74.5 | 74.4 | **73.5** | 80.1 | 76.6 |
| **EE-GL-a** | 71.0 | **89.3** | 79.1 | 71.7 | **81.3** | **76.2** | 71.4 | **84.5** | **77.4** |

Table 2: Results for the Equity Underweight type on the single-event and multi-event sets (human-annotated label).

randomly sampled documents from the training set and annotate event triggers.

We use the same human-annotated demonstrations for all EE methods.

### 4.1.2 Results and Analysis

**Main Results** We show our main experimental results in Table 1. We can observe that: (1) **ICL** achieves promising results (-7.7, +0.6, -4.1, -11.1 micro-F1 compared with **DE-PPN**) on four event types. Note that previous studies (Han et al., 2023; Wei et al., 2023) have shown that in-context learning performs poorly on other event extraction datasets. We suppose that the performance is better on this dataset because the financial disclosure documents are required to organize in a highly homogeneous format. This result indicates the power of in-context learning. (2) Both **GL-r** and **GL-a** outperform **ICL** on four event types by at most +2.9, +1.2, +3.3, +4.1 micro-F1. Note that we only use extra trigger labels of 50 documents per class. (3) Though out three-step methods and the summary agent can slightly improve the performance (**GL-b** vs. **ICL**), the main performance gain comes from the learned guidelines (**GL-r** vs. **GL-b**). (4) **GL-a** consistently outperforms **GL-r** by a small margin, which verifies the effectiveness of our active learn-

ing method. Note that **DE-PPN** is trained on 25631 fully annotated examples, while our methods are trained on 200 examples in total with only trigger annotation.

**Results on Human-Annotated Test Set** As the label constructed by distant supervision is noisy, we manually annotate the test set of *Equity Underweight*. The results on this test set are shown in Table 2. It shows that: (1) **GL-r** and **GL-a** improve 1.7, 2.5 F1 scores over **ICL**, respectively. (2) **ICL** and **GL-r/a** outperform **DE-PPN** by over 10% micro-F1. This implies that though only provided few manual labels, LLMs are more capable of aligning with human annotation than supervised methods trained on a large-scale weakly-supervised dataset. (3) Supervised method **DE-PPN** performs much poorer on multi-event documents than single-event document (53.4 vs. 79.3), while ICL-based methods are more robust (more discussion on Appendix A.1.4).

### 4.2 Relation Extraction

### 4.2.1 Setups

**Dataset** We use **SemEval 2010 task 8** (Hendrickx et al., 2010) relation extraction dataset to conduct our experiments. This task focuses on semantic relations (e.g., "component and container",

| Method | P. | R. | F1. |
|--------|-----|-----|-----|
| **RIFRE♣** | - | - | 91.3 |
| **RE-ICL** | 58.3 | 67.7 | 62.7 |
| **RE-GL-b** | 59.3 | 67.1 | 63.0 |
| **RE-GL-r** | 62.3 | 69.7 | 65.8 |
| **RE-GL-a** | **63.5** | **70.6** | **66.9** |

Table 3: Results on the SemEval dataset. ♣: SOTA supervised model (Zhao et al., 2021).

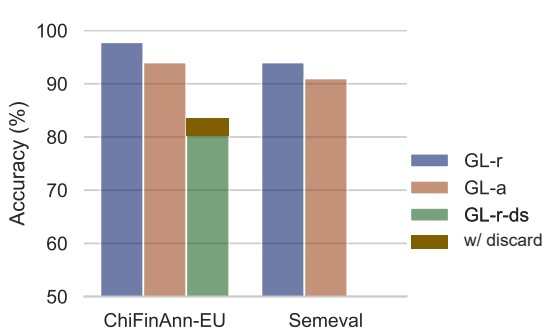

Figure 4: The manual evaluation results of the learned guidelines on ChFinAnn-EU (EE) and SemEval (RE) dataset (randomly select 50 for each evaluation).

"entity and destination") between pairs of nominals and contains 10,717 annotated examples covering nine relations collected from general domain resources. We randomly sample 1000 test samples from the original test set for evaluation.

**Method** We directly apply guideline learning to conduct the relation extraction task as detailed in Sec 3.2.

**Compared Models** (1) **RIFRE** (Zhao et al., 2021): SOTA supervised model. (1) **RE-ICL**: For a pair of entities in a text, we prompt the LLM to directly output the relation type. (3) **RE-GL-b**: Baseline version of our guideline learning method with empty guidelines. (4) **RE-GL-r**: Our guideline learning method. We *randomly* sample 500 instances (50 instances per relation class on average) from the training set to learn guidelines. (5) **RE-GL-a**: We *actively* sample 500 instances out of 1000 randomly sampled instances from the training set to learn guidelines.

#### 4.2.2 Results and Analysis

The results are shown in Table 3. We can observe that (1) **GL-r** and **GL-a** outperform **ICL** by 3.1, 4.2 F1-scores, respectively. This verifies the effectiveness of applying our guideline learning framework for relation extraction. (2) The performance of ICL-based RE is still far behind SOTA methods (66.9 vs.

91.3), which is consistent to previous studies (Han et al., 2023).

### 4.3 Analysis

#### 4.3.1 Quality Evaluation of Guidelines

We manually evaluate the quality of learned guidelines. Specifically, for each task, we randomly sample guidelines from the best epoch and compute the accuracy where we count a hit if the guideline is precise and unambiguous. The results are shown in Figure 4. For both **GL-r** and **GL-a**, which are provided manual labels, the accuracy is above 90%. This indicates that LLMs can well perform the generalizing task when appropriately prompted. To investigate how the label quality effects the quality of generated guideline, we conduct experiments (**GL-r-ds**) with the same setting as **GL-r** but providing the distant supervised labels. The accuracy drops dramatically by 17.2 points. The forgetting mechanism (**w/ discard**, detailed in Sec 2.3) helps to discard harmful guidelines boosting the accuracy by 3.3 points, but it is still significantly lower than **GL-r**. This indicating the necessity of label quality for generating high-quality guidelines.

#### 4.3.2 Case Study of Guidelines

Note that we generate guidelines by first generalizing the input instance to its general form, then combining it with its golden label. This implementation can successfully generate helpful guidelines, while inevitably makes some mistakes. We show some cases in Figure 5. We find some helpful guidelines imply annotation rules in the annotation guidelines (e.g., He-4). The cause of the harmful guidelines is mainly due to the inadequate generalization (e.g. Ha-1, Ha-3) and annotation error (e.g. Ha-2). Besides, in extreme cases, the relation between two entities is only based on the literal meaning of the entity (e.g. Ha-4), which is hard to generate a general guideline.

#### 4.3.3 Comparison with DE-PPN in Data Scarcity Settings

We conduct experiments to investigate how ICL-based approaches compare to alternative supervised approaches in settings where annotation is scarce. Specifically, we train DE-PPN on (1) the 192 annotated documents available to ICL approaches (50 documents per event type); (2) 5k annotated documents (random sampled); (3) all 29k annotated documents. We compare DE-PPN with vanilla few-shot ICL (EE-ICL) and our guideline

| | Helpful | Harmful |
|---|---|---|
| **EE** | **He-1.** The shares sold through the trading system trigger an equity underweight event. **He-2.** The freely tradable shares held before the reduction don't trigger an equity underweight event. | **Ha-1.** The shares bought and sold mistakenly triggers an equity underweight event. **Ha-2.** The outstanding shares sold through the centralized bidding trading system don't trigger an equity underweight event. |
| **RE** | **He-3.** "use the Y (Technology) to inform about X (Topic or Subject)" indicates that the relation between X and Y is MESSAGE AND TOPIC. **He-4.** "Y (Agriculture) was put inside the upper part of the rock X (Building)." indicates that the relation between X and Y is ENTITY AND DESTINATION. *(NOT CONTAIN AND CONTAINER because of the motion verbs prevailing over "stative" relations criteria.)* | **Ha-3.** "early in the Y (Product)" indicates that the relation between X and Y is MESSAGE AND TOPIC. **Ha-4.** "X (Food) Y (Food)" indicates that the relation between X and Y is ENTITY AND ORIGIN. *(The original sentence: Homemade tomato soup is so much better than the shop bought versions.)* |

Figure 5: Case study of guidelines learned in EE and RE task. We use colors for better illustration.

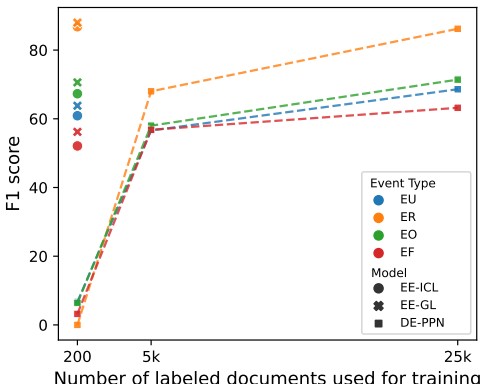

Figure 6: F1 scores of different methods trained on different training dataset sizes. We use different colors, markers to distinguish different event types and models, respectively.

learning approach (EE-GL) on the same test set. The F1 score of each event type is shown in Figure 6. We find that DE-PPN fails when only providing 192 labeled documents, with very low F1 scores on all event types. The problem is alleviated when providing 5k labeled documents. DE-PPN relies on a large amount of annotated data to work well. This indicates the superiority of ICL approaches over data-hungry supervised approaches. Our guideline learning approach further improves the few-shot ICL approach (EE-ICL) on all event types.

## 5 Related Work

### 5.1 In-Context Information Extraction

Information extraction (IE) extracts structured knowledge of interest from unstructured text, includes entities, relations between entities, event arguments, etc. Previous studies mainly focus on fine-tuning a task-specific model under the supervision from large-scale datasets (Zhao et al., 2021; Zheng et al., 2019; Yang et al., 2021; Liang et al., 2022). Though achieving remarkable performance, these models heavily rely on high-quality manually-

annotated datasets and may fail in new scenario.

On the other hand, Brown et al. (2020) shows that in-context learning (ICL) of large language models (LLMs) can perform numerous tasks when provided a few examples in a natural language prompt. ICL is a highly promising new learning paradigm because it is tuning-free, user-friendly, and data-efficient. There are many studies applying in-context learning to perform IE tasks. Wan et al. (2023) proposes GPT-RE to bridge the gap between ICL and finetuning baselines for RE via two strategies: entity-aware demonstration retrieval and gold-label induced reasoning. Chen et al. (2023) propose an in-context learning-based NER approach and model PLMs as a meta-function, which can inject in-context NER ability into PLMs and recognize entities of new types on-the-fly using only a few demonstrative instances. However, though focusing on ICL, these methods still requires training over large-scale datasets.

Recently, ChatGPT (OpenAI, 2023a) has stimulated the research boom in the field of LLMs. ChatGPT has been the most well-known and powerful LLM so far, with amazing ability of ICL and instruction following. There are many studies exploring ChatGPT's capability on IE tasks. Many studies (Han et al., 2023; Wei et al., 2023; Gao et al., 2023) evaluate ChatGPT's capability on IE tasks by directly prompting and find a huge performance gap between ChatGPT and SOTA results. They mainly focus on performance evaluation without in-depth investigations to boost ICL ability for IE tasks.

### 5.2 Retrieval-augmented ICL

Many studies propose to retrieve relevant evidence from extra knowledge sources to enhance the performance of ICL. Demonstration retrieval aims at designing more effective strategies for judiciously selecting in-context examples from a large training

set. For example, Liu et al. (2022) applies kNN-retrieval based on sentence-level representations. GPT-RE (Wan et al., 2023) further finetunes an entity-aware representation on the training set for better retrieval. However, similar to the supervised paradigm, these methods still rely on a large-scale annotated dataset. Some studies retrieve relevant information from an extra memory to assist in ICL. Madaan et al. (2022) proposes a memory-assisted framework that correct errors via user interactions. They pair the GPT-3 (Brown et al., 2020) with a growing memory of recorded cases and user feedback, which allows the system to produce enhanced prompts for any new query. However, their method heavily replies on the quality of user interaction. As they use simulated user feedback in experiments, the effectiveness and stability have not been verified in real-world cases.

Our approach utilizes similar memory and retrieval mechanism. With a focus on IE, our framework can automatically learn high-quality guidelines from few error cases, obviating the need for user feedback, which is more efficient and stable.

## 5.3 Instruction Learning

Guideline Learning differs from two main branches of previous work on instruction learning:

**Instruction induction via ICL**. Honovich et al. (2023) predict the task instruction by prompting instruction-tuned LLMs. They conduct explorative experiments, focusing on tasks that have "clear and simple instructions". In contrast, our GL framework focuses on more complex instructions with a highlight on IE tasks: extraction of complex concepts. We propose the "guideline" as a bridge to learn and utilize more specific instructions from error cases automatically, which can be viewed as an in-depth extension of previous work.

**Instruction learning for meta-training**. Ye et al. (2023) propose to utilize instruction learning to better finetune LLMs and boost the zero-shot performance. Our GL framework aims at boosting the model performance under the tuning-free setting, which is orthogonal to their work.

## 6   Conclusion

This paper explores the underspecified task description problem in in-context information extraction. We propose a guideline learning framework to alleviate the problem, which automatically learns guidelines from few labeled instances during the learning phrase, and retrieving helpful guidelines to assist in reasoning during inference. Our experiments on event and relation extraction show that a straightforward implementation of guideline learning can enhance vanilla in-context learning by approximately 4%.

## Limitations

The guideline learning (GL) framework establishes a powerful and reproducible starting point for in-context learning research. However, our work still lacks depth in certain aspects and many potential research directions within this framework warrant further investigation.

**Broader applications**   In this paper, we only apply GL to IE tasks to alleviate the *underspecified task description* problem. It's encouraging to transfer GL to other tasks with complicated task specifications.

**More specialized retriever**   We implement an elementary retriever by utilizing OpenAI's embedding API. Though sufficient to verify the effectiveness of our framework, the performance is suboptimal. It's promising to establish a more powerful retriever that specializes in retrieving relevant guidelines based on input cases.

**More sophisticated generalizer**   We generate guidelines by prompting an LLM agent to properly extrapolate each error case. The guidelines are mostly precise but still lack generality. It's possible to design a more sophisticated generalizer to summarize a guideline based on multiple similar error cases.

**Enhance the rule-following capability of LLMs** One key necessary capability of the reasoner is to generate responses while faithfully following input rules. We observe that gpt-3.5-turbo, the backbone LLM agent in our experiments, still struggles to truly refer to relevant rules. We present a preliminary discussion in Appendix A.4. It would be intriguing to evaluate and enhance the rule-following ability of LLMs.

## Acknowledgements

This work has been supported by the National Natural Science Foundation of China (No. 62076231, 62206265), and the China Postdoctoral Science Foundation (No. 2021M703271). We thank all the anonymous reviewers for their valuable and constructive comments.

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

## A Appendix

### A.1 Event Extraction Experiment Details

#### A.1.1 ChFinAnn Dataset

The ChFinAnn dataset (Zheng et al., 2019) is constructed from real-world Chinese financial documents via event-level distant supervision. It contains 32040 documents in total, focusing on five event types: Equity Freeze (EF), Equity Repurchase (ER), Equity Underweight (EU), Equity Overweight (EO) and Equity Pledge (EP). We don't conduct experiments on EP events as we suppose there exists event confusion that both equity pledge and release of pledge are labeled as EP events. As the official API (gpt-3.5-turbo-0301) has a max token length of 4096 tokens, we only keep the documents with a length less than 1000 Chinese characters. We sample at most 200 documents for each event type from these documents. Table 4 presents the data statistics.

We calculate the ratio of negative triggers (i.e. candidate shares that refer to non-events) in each event type on our test set. The results are shown in Table 5. The ratio of negative triggers varies across different event types, ranging from a minimum of 23.1% to a maximum of 63.9%. The simple trigger expression "number shares" we use for this dataset ensures high recall (every event record on this dataset involves such expression), however, it also introduces unnecessary negative triggers, resulting in additional cost of event trigger classification. This indicates that identifying and classifying triggers on this dataset is non-trivial. Note that our experiments are designed to validate the GL framework, with a focus on trigger classification. Consequently, we do not place much emphasis on trigger identification. In practice, it's more efficient to design a powerful event trigger identifier beyond the simple pattern. For example, it's promising to prompt the LLM to identify candidate triggers with few in-context demonstrations. We leave it as future work.

| Event | # Test | # Our Test | Ratio |
|-------|--------|-----------|-------|
| EF | 204 | 174 | 85.3% |
| ER | 282 | 200 | 70.9% |
| EU | 346 | 193 | 55.8% |
| EO | 1138 | 165 | 14.5% |

Table 4: Dataset statistics about the number of documents for the test set (# Test) and the test set in our experiments (# Our Test).

| Event | # Candidate | # Negative | Ratio |
|-------|-------------|-----------|-------|
| EU | 790 | 468 | 59.2% |
| ER | 260 | 60 | 23.1% |
| EO | 534 | 341 | 63.9% |
| EF | 477 | 262 | 54.9% |

Table 5: Dataset statistics about the number of candidate triggers and negative triggers in our test set.

#### A.1.2 Prompts

For guideline learning, we conduct binary classification for each event type. We present the prompt of the equity underweight events. Only the demonstrations in the prompt are different across different event types. We use 6-8 demonstrations for each LLM agent. We introduce our method in section 4.1.1. Here we briefly recap the input and output of each LLM agent:

1. The summarizer takes a document and one share in it as the input and output a summary of this share, which we call share description. The prompt is presented in Figure 7.

2. The generalizer takes the instance (share description) as input and output its general form by abstracting common properties. The prompt is presented in Figure 8.

3. The reasoner takes the instance (share description) and the retrieved guidelines as input and output the reasoning process (CoT), the predicted answer and the index of the used guideline. The prompt is presented in Figure 9.

For EAE, we prompt the LLM to output the event table in the markdown format. As the documents in this dataset are long, we only use 2 demonstrations in each prompt. **EE-ICL** and **EE-GL** use the same task instruction and demonstrations. The only difference is that **EE-GL** provides the candidate trigger shares identified by the ETC method. The prompts are presented in Figure 10 and Figure 11.

#### A.1.3 Hyper-parameters

Note that for the reasoner, we apply Self-Consistent Chain-of-Thoughts (SC-CoT) prompting (Wang et al., 2022b). We show the hyper-parameter settings in Table 6. For EAE, we use a very low temperature 0 to generate stable outputs.

Figure 7: The prompt (translated) of the summarizer for the *equity underweight* event (ChFinAnn dataset). The {document} and {share} denotes the input document and share, respectively.

| Module | Hyper-parameter | Value |
|---|---|---|
| | the number of epochs | 5 |
| **Recall** | the maximum number of retrieved guidelines | 3 |
| | retrieval threshold | 0.95 |
| **Reason** | SC-CoT trials | 8 |
| | SC-CoT sampling temperature | 1 |
| **Reflect** | the score threshold to discard a harmful guideline | 0 |

Table 6: Hyper-parameter settings for event extraction.

## A.1.4 Discussion on Single-F1 vs. Multi-F1

More precisely, "Multi" denotes "multi-record" rather than "multi-event", which means there are multiple records for one event type in a document. The fact that multi-record performance is lower than single-record performance is widely observed among all supervised approaches on this dataset. For example, Doc2EDAG (Zheng et al., 2019): single 82.3 vs. multi 67.3, GIT (Xu et al., 2021): single 87.6 vs. multi 72.3 (averaged F1 scores). One possible reason is data imbalance (only 28.5% of all documents contain multi-record events). Another possible reason is the difficulty of multi-record documents. It's interesting that ICL approaches seem to be more robust against the number of records in documents than supervised approaches. Specifically, the gap between single-F1 and multi-F1 is relatively low for Guideline Learning, as shown in Table 2. This is out of the scope of this paper and we leave it as future work.

## A.2 Relation Extraction Experiment Details

### A.2.1 Prompts

For guideline learning, we directly conduct multi-class relation classification. There are two main components:

1. The generalizer takes the instance (a sentence and one entity pair) as input and output the general form. This is decomposed into two steps: extracting relevant text pieces and abstract entity types. The prompt is presented in Figure 12. The generalizer combines the two responses (the text span and entity types) to get the final general form.

2. The reasoner takes the instance (a sentence and one entity pair) and the retrieved guidelines as input and output the reasoning process, the predicted answer and the index of the used guideline. The prompt is presented in Figure 13.

Figure 8: The prompt (translated) of the generalizer for the *equity underweight* event (ChFinAnn dataset). The {description} denotes the input share description.

For **RE-ICL**, we apply chain-of-thought prompting. The prompt is presented in Figure 14. We use 10 demonstrations for the reasoner and **RE-ICL**.

### A.2.2 Hyper-parameters

Note that for the reasoner and **RE-ICL**, we apply Self-Consistent Chain-of-Thoughts (SC-CoT) prompting (Wang et al., 2022b). We show the hyper-parameter settings in Table 7.

### A.3 Discussion on Generalizer

The intuition of the generalizer is in two folds. First, the guideline should have some generalizability to cover/handle similar cases. Secondly, in practice, the generalizer is helpful to the guideline retrieval task based on the input case. If the guidelines are composed of corrected error cases, the retrieval would be case-to-case, which is very sensitive. For example in the following quote block, the input case is more similar to G1 literally. However, G2 is more relevant as they both describe an active underweight event. If we generate their general form by abstracting common properties (company name, number of shares, date), it will be more similar to G2.

> **Input case**: Xinguang Investment actively reduced its shareholdings of this company by 300,000 shares.
>
> **G1**: Xinguang Investment passively reduced its shareholding in the company by 300,000 shares. This does not trigger an EU event.

> **G2**: Jinying Technology actively reduced its shareholdings of this company by 200,000 shares today. This triggers an EU event.
>
> **General form of input case**: One company actively reduced its shareholdings of another company.
>
> **General form of G1**: One company passively reduced its shareholdings of another company. This does not trigger an EU event.
>
> **General form of G2**: One company actively reduced its shareholdings of another company. This triggers an EU event.

In our experiments, we implement the generalizer by few-shot prompting an LLM agent. Though the generalizer is critical for the GL framework, we don't put the implementation details into section 3 as there may be other underlying implementations, for example, finetuning a more effective generalizer, and we want to highlight our contribution on the guideline learning framework itself.

### A.4 Discussion on Rule-following Capabilities

We manually evaluate the following aspects of responses from Reasoner: 1. **relevant**: whether the rules referred to by the model are truly relevant to the instance; 2. **well-referred**: whether the model genuinely follows the rules, i.e. the response is consistent with the rules it refers to. We analyze 50 responses generated by the gpt-3.5-turbo and gpt-4

---

**EE - Reasoner**

**Please determine whether the number of shares meets the given statement based on the share description and guidelines. If you need to refer to a guideline, please refer to the one that is most relevant to the share description. If there is no guideline to refer to, please make a direct judgment. The output format is {"reasoning": "reasoning process", "used_guideline": "index of the used guideline", "answer": "yes or no"}.**

-------------------------------------------------------------------------------------------------------------------------------------

**Share description**: 168,700 shares are the number of shares that Mr. Huang Shengqiu, the actual controller of the company, reduced on November 1, 2018.
**Statement**: The shares are reduced by an equity holder in an equity underweight event.
**Guideline**: None.
**Answer**: {"reasoning": "According to the share description, it is the number of shares that Mr. Huang Shengqiu has reduced. Without any guideline, I can only make a direct judgment. The shares reduced by the shareholder (Mr. Huang Shengqiu) meets the statement, therefore the answer is yes.", "used_guideline": "-1", "answer": "yes"}

-------------------------------------------------------------------------------------------------------------------------------------

**Share description**: 464,900 shares are the number of shares disposed of by Mr. Peng Xunde due to default on stock pledge on December 25, 2018.
**Statement**: The shares are reduced by an equity holder in an equity underweight event.
**Guideline**: 1. The shares disposed of due to default in stock pledge meets the statements. 2. The shares held before passive reduction does not meet the statements. 3. The number of shares held after reduction does not meet the statements.
**Answer**: {"reasoning": "According to the share description, it is the number of shares disposed of due to Mr. Peng Xunde's default on stock pledge. The most relevant guideline for this description is guideline 1. The shares disposed of due to the default on stock pledge by the shareholder (Mr. Peng Xunde) meets the statement, therefore the answer is yes.", "used_guideline": "1", "answer": "yes"}

-------------------------------------------------------------------------------------------------------------------------------------

**more demonstrations...**

-------------------------------------------------------------------------------------------------------------------------------------

**Share description**: {description}
**Statement**: The shares are reduced by an equity holder in an equity underweight event.
**Guideline**: {retrieved_guidelines}
**Answer**:

---

Figure 9: The prompt (translated) of the reasoner for the *equity underweight* event (ChFinAnn dataset). The {description} denotes the input share description. The `{retrieved_guidelines}` denotes the guidelines retrieved from the knowledge base.

agents[4]. The results (accuracy) are shown in Table 8. We find that gpt-3.5-turbo is capable of figuring out and following relevant rules, while gpt-4, known as the most powerful LLM, makes fewer mistakes. We utilize gpt-3.5-turbo as the backbone LLM in our experiments, which is sufficient to verify our framework. Moreover, our framework may potentially gain additional advantages from the increasing rule-following capabilities of these backbone LLMs.

---

[4]gpt-3.5-turbo-0301 and gpt-4-0613

Figure 10: The prompt (translated) of the **EE-ICL** for the *equity underweight* event (ChFinAnn dataset). The {document} denotes the input document.

Figure 11: The prompt (translated) of the **EE-GL** for the *equity underweight* event (ChFinAnn dataset). The {document} and {shares} denotes the input document and candidate trigger shares identified by previous ETC methods, respectively.

> **RE - Generalizer - 1**
>
> **Find the text span in the quoted sentence that may indicate the relation between two entities. Remove irrelevant words in the text span and make sure your answer is only the text span.**
>
> ---------------------------------------------------------------------------------------------------------------------------------
>
> **Context**: These city dwellers have sunk into abominations, after the rain.
>
> **Entities**: city dwellers and abominations
>
> **Answer**: city dwellers have sunk into abominations
>
> ---------------------------------------------------------------------------------------------------------------------------------
>
> **more demonstrations...**
>
> ---------------------------------------------------------------------------------------------------------------------------------
>
> **Context**: {sentence}
>
> **Entities**: {entities}
>
> **Answer**:

> **RE – Generalizer - 2**
>
> **You are an NLP expert. You are knowledgeable in taxonomy. Please tell me the category of an entity. Note that the category should be general and precise. For example, the following category is good: Person, Location, Organization, Event, Product, Action, Time. Your answer should only contain one word or phrase.**
>
> **Sentence**: {sentence}
>
> The category of {entities} is:

Figure 12: The prompt of the generalizer in RE. The {sentence} and {entities} denotes the input sentence and the entity pair, respectively.

> **RE - reasoner**
>
> **You are a knowledgeable person. You will solve the relation extraction task. Given the context, you will first consider whether the most precise relation between two entities belongs to the following nine possible relations. If yes, you will output the most precise relation, otherwise you will output NULL:**
>
> **CAUSE AND EFFECT: X and Y have a CAUSE AND EFFECT relation if X is an event or object that leads to an effect Y.**
>
> **INSTRUMENT AND AGENCY: X and Y have an INSTRUMENT AND AGENCY relation if X is an agent that uses an instrument Y.**
>
> **PRODUCT AND PRODUCER: X and Y have a PRODUCT AND PRODUCER relation if X is a producer that causes a product Y to exist.**
>
> **… more relations**
>
> **The output format should be {{"reasoning": "my reasoning process", "used_guideline": "the index of the guideline that you used to answer the question", "answer": "the most precise relation"}}**
>
> **Demonstration:**
>
> ---------------------------------------------------------------------------------------------------------------------------------
>
> **Context**: The apple blossom season usually runs from mid-april to early may.
>
> **Entities**: apple and blossom
>
> **Guideline**: None.
>
> **Answer**: {"reasoning": "According to the meaning of the Context, the 'blossom' is a component of an apple tree. Therefore, the most precise relation between 'apple' and 'blossom' is COMPONENT AND WHOLE.", "used_guideline": "-1", "answer": "PRODUCT AND PRODUCER"}
>
> ---------------------------------------------------------------------------------------------------------------------------------
>
> **Context**: public brand products were donated to charities .
>
> **Entities**: brand and charities
>
> **Guideline**: 1. "X is donoted to Y" indicates that the relation between X and Y is ENTITY AND DESTINATION. 2. "X caused by Y" indicates that the relation between X and Y is CAUSE AND EFFECT.
>
> **Answer**: {"reasoning": "The context indicates that the brand (X) is the entity that is being donated and the charities (Y) are the destination towards which the brand products are being donated. Therefore, the most precise relation between 'brand' and 'charities' is ENTITY AND DESTINATION.", "used_guideline": "1", "answer": "ENTITY AND DESTINATION"}
>
> ---------------------------------------------------------------------------------------------------------------------------------
>
> **more demonstrations...**
>
> ---------------------------------------------------------------------------------------------------------------------------------
>
> **Context**: {sentence}
>
> **Entities**: {entities}
>
> **Guideline**: {retrieved_guidelines}
>
> **Answer**:

Figure 13: The prompt of the reasoner in RE. The {sentence}, {entities}, and {retrieved_guidelines} denotes the input sentence, the entity pair, and the retrieved guidelines, respectively.

Figure 14: The prompt of the **RE-ICL** in RE. The {sentence} and {entities} denotes the input sentence and the entity pair, respectively.

| Module | Hyper-parameter | Value |
|---|---|---|
| | the number of epochs | 3 |
| **Recall** | the maximum number of retrieved guidelines
retrieval threshold | 3
0.92 |
| **Reason** | SC-CoT trials
SC-CoT sampling temperature | 5
1 |
| **Reflect** | the score threshold to discard a harmful guideline | 0 |

Table 7: Hyper-parameter settings for event extraction.

| Model | Relevant | Well-Referred |
|---|---|---|
| gpt-3.5-turbo | 0.84 | 0.88 |
| gpt-4 | **0.94** | **0.98** |

Table 8: Manual evaluation of rule following capabilities.