# OpenReview forum: "Guideline Learning for In-Context Information Extraction"
_EMNLP/2023/Conference — EMNLP 2023 Main_

### Official Review · Reviewer_NFNq · 2023-08-04

**Typos Grammar Style And Presentation Improvements:** L075
**Soundness:** 4

**Excitement:**

4: Strong: This paper deepens the understanding of some phenomenon or lowers the barriers to an existing research direction.

**Paper Topic And Main Contributions:**

This paper proposes a Guideline Learning (GL) framework for improving the in-context learning (ICL) in information extraction (IE) tasks.
The authors identify a challenge in ICL, namely the conceptual bias, where the "comprehended concept by LLMs might diverge from the target concept". To address this, they introduce a GL framework that learns to generate guidelines from a few annotations during the learning phase, and retrieves helpful guidelines for better ICL during inference.

The authors demonstrate the effectiveness of their approach through experiments on event extraction and relation extraction tasks. The results show the GL framework can enhance ICL, although it still falls behind SOTA supervised models in general. The authors also manually labeled a test set (Equity Underweight), and found their approach outperforms supervised models in this particular case.

**Reasons To Accept:**

1. GL is an interesting and promising direction in the era of LLMs. This paper proposes an innovative and interesting approach.
2. Results suggest this framework is effective, compared to the vanilla ICL method.
3. The error analysis and case study can provide some insights.

**Reasons To Reject:**

1. GPT-3 is known to give unstable outputs, although that is more of an issue in text generation. The evaluation probably needs multiple runs, or at least addresses why it is not needed.
2. Only one dataset is used, for the two tasks.

**Reproducibility:**

3: Could reproduce the results with some difficulty. The settings of parameters are underspecified or subjectively determined; the training/evaluation data are not widely available.

**Reviewer Confidence:**

4: Quite sure. I tried to check the important points carefully. It's unlikely, though conceivable, that I missed something that should affect my ratings.

---

> ### Author Rebuttal · Authors · 2023-08-28
>
> Thanks for your positive feedback! We are glad that you think our GL approach is innovative and effective, and the error analysis and case study can provide some insights. In the following, we address your concerns one by one.
>
> **Q: The evaluation probably needs multiple runs, or at least addresses why it is not needed.**
>
> A: Thanks for your suggestion. We have set a very low temperature in our experiments, which makes the responses more focused and deterministic. We only do a single run for each experiment to lower our cost. We will also report the variance in our next revision.
>
> **Q: Only one dataset is used, for the two tasks.**
>
> A: Though we only conduct experiments on one dataset for EE and RE task, our framework can be easily transferred to other datasets, with simple modifications on prompts. We would like to apply the GL framework to other tasks in the future.
>
> **If you have any further questions or concerns, please do not hesitate to inform us.**
>
> **Best regards.**

---

### Official Review · Reviewer_B35m · 2023-08-04

**Soundness:** 3

**Excitement:**

4: Strong: This paper deepens the understanding of some phenomenon or lowers the barriers to an existing research direction.

**Paper Topic And Main Contributions:**

The paper addresses the problem of information extraction (IE),
specifically event and relation extraction, through in-context learning
using instruction-tuned large language models, specifically GPT3.
Noting mounting evidence that the strength these models exhibit for
other NLP tasks does not necessarily extend to IE, the paper argues
that this is due to insufficient context for correct performance of
these relatively complex tasks.  It therefore proposes an approach
called "guideline learning," a pre-training phase in which the models
generate elaborations of the guidelines in an error-driven fashion
using a modest number of labeled examples.  At test time, the
guidelines, including the generated ones, as well as illustrative
training examples, are retrieved to form part of the context used to
prime the model for a particular example.  In experiments involving
event classification in a Chinese financial domain and relation
classification on a SemEval problem set, the paper demonstrates
consistent improvement over a generic in-context learning framing
using the same model, though performance lags behind fully supervised
baselines trained on many more examples.

**Questions For The Authors:**

A. I didn't really understand the observation that DE-PPN perform poorly
on multi-event documents ("due to data imbalance").  Please explain.

B. Table 4 in the appendix is useful, but doesn't answer a question I had
in assessing the event classification results.  The trigger expression
seems very inclusive.  How often does it nominate an expression that
matches none of the classes you targeted?

**Reasons To Accept:**

In an area of active research and considerable interest to the NLP
community, the paper proposes an interesting approach and demonstrates
its value convincingly.  The work will be of interest to practitioners
attempting to address IE problems where annotated data is in short
supply.  Because IE admits so many variants, such problems are not
hard to come by.  Thus, the practical contribution of the paper is
real.

The paper is well written and mostly clear.  The related work section
is better written than in many papers, with a nice focus that helps
orient the reader to this relatively active research area.

The idea pursued in the paper is intuitive and described with
sufficient clarity that the work should be eminently replicable.  The
paper's appendices include ample information useful for other
researchers seeking to build on this work.

**Reasons To Reject:**

It's not clear how significant this work ultimately is.  Yes, it
appears to improve the SOTA in the area of in-context learning for IE,
but performance improvements over generic ICL, although consistent and
apparently significant, do not really approach those of the fully
supervised approaches.  In addition, it's not clear how this approach
compares to alternative supervised approaches in a setting where
annotation is scarce.  The evaluated supervised baselines were all
trained on all available training data. It would be useful to see what
performance levels they achieve when limited to the 500 examples
available to the ICL approaches.

The phrase "conceptual bias" is decidedly vague and doesn't really
seem to correspond to the definition the paper gives it.  The problem
the paper targets is that the tasks we're asking the model to perform
are chronically underspecified, because it's difficult to describe
them succinctly.  It's clear that this is a problem that impacts
inter-annotator agreement in human annotation, so the idea that it
negatively impacts ICL is intuitive, and the paper's idea appealing.
However, the paper makes the rather strong claim that "the suboptimal
performance of in-context IE is attributable to...conceptual bias."
The paper does not substantiate this claim.  It shows that this kind
of "guideline learning" provides lift, but the resulting models are
presumably still suboptimal (e.g., they presumably lag behind human
performance).  Bottom line: I think a little more modesty in the
framing of the work would be good.

After admirable presentation of the approach and its motivation, the
experiments underwhelm in certain respects.  First, the approach
includes a focused summarization step in response to limited GPT3
input buffers that is not part of the retrieve-reason-reflect
framework articulated in the paper.  While understandable, this
expedient has an unclear impact on performance.  Second, the RE
experiments randomly sample test examples from the SemEval test set,
complicating comparison with reported SemEval results and making it
hard for follow-on research to replicate the paper's results.
Finally, despite the paper's framing of IE as complex, the financial
IE task seems relatively simple--perhaps one reason it is shown to be
competitive with DE-PPN in the experiments reported in Table 2.  In
contrast to the typical event extraction problem, where there are a
large number of possible event types, this problem has four types.  In
conventional event classification, triggers may refer to non-events.
It's not clear from the paper's explication whether and how much this
is a feature of the problem the paper addresses.

**Reproducibility:**

4: Could mostly reproduce the results, but there may be some variation because of sample variance or minor variations in their interpretation of the protocol or method.

**Reviewer Confidence:**

4: Quite sure. I tried to check the important points carefully. It's unlikely, though conceivable, that I missed something that should affect my ratings.

**Typos Grammar Style And Presentation Improvements:**

The English in the paper is very good, but it had difficulty with
number agreement between subject and verb in a number of places.  A
native English speaker could find and correct these things quickly.
However, these grammatical discrepancies never damaged the clarity of
your explication.

You address limitations of the approach in the conclusion, and the
limitations you cite seem thoughtful.  However, it's more typical to
have a separate Limitations section.  I believe such a section does
not count toward the page budget, so moving this discussion to such a
section would both help the reader and buy you more space.

---

> ### Author Rebuttal · Authors · 2023-08-28
>
> Thanks for your detailed comments. We are glad that you like our presentation of the approach and its motivation. We agree that the phrase "conceptual bias" is imprecise and some claims are too strong. We would like to use the phrase "underspecified task description" instead. Besides, it seems that you are mostly concerned about some aspects of our experiments. We conduct some additional experiments for them. In the following, your comments are first stated and then followed by our point-by-point responses.
>
> **Q: It's not clear how this approach compares to alternative supervised approaches in a setting where annotation is scarce.**
>
> A: We conduct an experiment to evaluate the supervised model DE-PPN in the data-scarcity setting. Specifically, we train DE-PPN on (1) the **192 annotated documents** available to ICL approaches (50 documents per event type); (2) **5k annotated documents** (random sampled); (3) **all 29k annotated documents**. We use 20% of the data as the dev set. We compare several models on the same test set in our paper: 1. **DE-PPN**, 2. vanilla few-shot ICL (**EE-ICL**), 3. Our guideline learning approach (**EE-GL**). The F1 score of each event type is shown below:
>
> | model | EU | ER | EO | EF |
> | --- | --- | --- | --- | --- |
> | EE-ICL_192 | 60.9 | 86.8 | 67.3 | 52.1 |
> | EE-GL_192 | 63.8 | **88.0** | 70.6 | 56.2 |
> | DE-PPN_192 | 6.7 | 0.0 | 6.4 | 3.2 |
> | DE-PPN_5k | 56.6 | 68.0 | 58.0 | 56.8 |
> | DE-PPN_29k | **68.6** | 86.2 | **71.4** | **63.2** |
>
> We find that DE-PPN fails when only providing 192 labeled documents, with very low F1 scores on all event types. The problem is alleviated when providing 5k labeled documents. DE-PPN relies on a large amount of annotated data to work well. This indicates the superiority of ICL approaches over data-hungry supervised approaches. Our guideline learning approach further improves the few-shot ICL approach (EE-ICL) on all event types.
>
> **Q: The approach includes a focused summarization step in response to limited GPT input buffers. While understandable, this expedient has an unclear impact on performance.**
>
> A: For event extraction (EE), we introduce two expedient changes: (1) we decompose EE into three sub-tasks: trigger identification, trigger classification, and argument extraction; (2) we introduce a summarization step to fit the limited context length. In order to exclude the influence of these factors on the performance, we use the exact same experimental setup as the GL method but do not provide any guideline to the reasoner, denoted as **EE-GL-w/o. gd**. The F1 score of each event type is shown below:
>
> | setting | model | EU | ER | EO | EF |
> | --- | --- | --- | --- | --- | --- |
> |1-step | EE-ICL | 60.9 | 86.8 | 67.3 | 52.1 |
> |3-step + summarization | EE-GL-w/o. gd. | 61.5 | 87.1 | 67.7 | 52.5 |
> |3-step + summarization | EE-GL-r | 63.4 | 87.9 | 70.1 | 53.9 |
> |3-step + summarization | EE-GL-a | **63.8** | **88.0** | **70.6** | **56.2**|
>
> We find that though the two expedient changes can slightly improve the performance (0.6, 0.3, 0.4, 0.4) compared to EE-ICL, **the main performance gain comes from the learned guidelines (2.3, 0.9, 2.9, 3.7).**
>
> **Q:  The financial EE task seems relatively simple. However, the typical event extraction problem features: (1)  a large number of possible event types; (2) triggers that may refer to non-events.**
>
> A: In this paper, we decompose EE into three subtasks to apply our GL framework, with a highlight on the event trigger classification. When there are a large number of possible event types, the reasoner can't classify all event types in a single session, since it's hard to concretely describe all event types due to the limited input length. In our implementation, we decompose the multi-class event trigger classification into multiple binary classifications for each class. Specifically, we prompt the reasoner (LLM agent) to binarily classify whether the input instance triggers a specific event type. This alleviates the challenge at the cost of more rounds of sessions. After the trigger classification, we extract event records of each event type, given the triggers classified to this event type. The triggers that refer to non-events will not be provided in the next step.
> **To sum up, our implementation is capable of addressing the two typical challenges of event extraction without any modifications.** However, as our paper focuses on addressing the "underspecified task description" problem via guideline learning, these features are not highlighted. The cost of trigger classification is still relatively high, which calls for further research on efficient classification for a large number of event types.
>
> **Q: I didn't really understand the observation that DE-PPN performs poorly on multi-event documents ("due to data imbalance"). Please explain.**
>
> A: More precisely, the term "multi-event" should be "multi-record", which means there are multiple records for one event type in a document. The fact that multi-record performance is lower than single-record performance is widely observed among all supervised approaches on this dataset. For example, Doc2EDAG [1]: single 82.3 vs. multi 67.3, GIT [2]: single 87.6 vs. multi 72.3 (averaged F1 scores). One possible reason is data imbalance (only 28.5% of all documents contain multi-record events).  Another possible reason is the difficulty of multi-record documents. It's interesting that ICL approaches seem to be more robust against the number of records in documents than supervised approaches. Specifically, the gap between the "single-F1" and "multi-F1" is relatively low for Guideline Learning, as shown in Table 2. This is out of the scope of this paper and we leave it as future work.
>
> **Q: The RE experiments randomly sample test examples from the SemEval test set, complicating comparison with reported SemEval results and making it hard for follow-up research to replicate the paper's results.**
>
> A: We don't test the model on the whole SemEval test set in the current version due to limited funds to access the OpenAI API. We will make our test set publicly available for comparison and replication.
>
> Additionally, we are having difficulty following your question B. Specifically, what is "trigger expression"?
>
> **We've appended our Limitations section below our rebuttal to Reviewer 6ZCo. If you have any further questions or concerns, please do not hesitate to inform us.**
>
> **Best regards.**
>
> [1] Doc2EDAG: An End-to-End Document-level Framework for Chinese Financial Event Extraction. EMNLP 2019.
>
> [2] Document-level Event Extraction via Heterogeneous Graph-based Interaction Model with a Tracker. ACL 2021.

---

### Official Review · Reviewer_6ZCo · 2023-08-05

**Soundness:** 3

**Excitement:**

4: Strong: This paper deepens the understanding of some phenomenon or lowers the barriers to an existing research direction.

**Missing References:**

Missing related work on instruction tuning, especially using the model to generate instructions, which is similar to generating guidelines.
To list a few:
- Honovich, Or et al. “Instruction Induction: From Few Examples to Natural Language Task Descriptions.” Annual Meeting of the Association for Computational Linguistics (2022).
- Ye, Seonghyeon et al. “Guess the Instruction! Flipped Learning Makes Language Models Stronger Zero-Shot Learners.” ICLR (2023)

**Paper Topic And Main Contributions:**

This paper introduces a new paradigm of guideline learning (or more specifically guideline refinement) that can improve in-context learning. Given a few training examples and a guideline consisting of a set of rules, the model will update the rules for wrongly-predicted examples. Then the updated rules will be added into the context for model inference.
The paper evaluates this framework on the classification component of event extraction and relation extraction tasks and shows that it improves performance.

**Questions For The Authors:**

A. Have the authors thought about the faithfulness of the rules returned by the Reasoner? In Lines 232-233, it is mentioned that R* is the set of rules that "the LLM refers to as being helpful during reasoning". Are these rules truly relevant to the instance and are they actually used by the LLM?

B. ~~Because the rule scoring and rule update rely on the true label y, instance selection can only be done over the instances that have labels.  This might be ok when compared with fully-supervised models, but the authors should note that this is not true few-shot learning (where the assumption is that few-shot examples are provided as input and the model does not have access to any other labels)~~

C. The generalizer is only mentioned in Section 4.3 and not in the methodology part (Section 3).  Can you provide some intuition on why the generalizer is needed or show some results that ablate this component? If the generalizer is critical for the model to work, I'd suggest moving it into Section 3.

D. In Table 2, it seems that guideline learning is not helpful for multi-event documents (75.0 F1 for ICL vs 74.4 F1 for GL), can the authors explain this behavior?

E. Can the authors think of a way to apply this framework to datasets that do not have readily available guidelines?

**Reasons To Accept:**

- Overall, this guideline learning framework is interesting and meaningful as it does not require fine-tuning the entire model. Although the paper only applies the idea to EE and RE, the idea is transferrable to a wide range of tasks.

**Reasons To Reject:**

- **This paper does not include the mandatory Limitation section, and thus should be desk-rejected.**
- The idea is related to instruction learning/induction, which aims to automatically generate or rewrite the instruction of the task. The guideline in this paper can also be viewed as part of the task instruction, the difference being only a subset of the rules are used for each instance.
- Few datasets have rule-based guidelines (this might be the reason why not the paper does not include results for more popular RE datasets such as TACRED / DocRED), which limits the application of the framework.
- Since some of the experiment settings (EE-GL-r, RE-GL-r) rely on randomly sampling in-context examples, the variance should be reported for results.

**Reproducibility:**

3: Could reproduce the results with some difficulty. The settings of parameters are underspecified or subjectively determined; the training/evaluation data are not widely available.

**Reviewer Confidence:**

4: Quite sure. I tried to check the important points carefully. It's unlikely, though conceivable, that I missed something that should affect my ratings.

**Typos Grammar Style And Presentation Improvements:**

Presentation improvements:
- - It would be beneficial to accompany Section 3 with some examples of the prompt and input/output. The current writing is very abstract. For example, can you give an example of a guideline before and after the update?


Typos:
- Line 48: fine-tuning LLM faces with the .... issues  -> fine-tuning LLM faces ... issues
- Line 57: promising performances -> promising performance

- Line 95: different with -> different from
- Line 374: as all the event is related to -> as all event types are related to
- Line 377: extra "we apply guide"

---

> ### Author Rebuttal · Authors · 2023-08-28
>
> Thank you for your constructive comments, and they are exceedingly helpful for us to improve our paper. We feel sorry to not include the mandatory Limitations section, but we do address the limitations of our approach in the conclusion (as reviewer B35m summarized). We have appended Limitations section at the end of this comment and will add it to our next revision. It appears there are some misunderstandings regarding our framework, and one important thing we want to highlight is that our model **does NOT** need annotated guidelines.
>
> In the following, we address your concerns one by one.
>
> **Q: The idea is related to instruction learning/induction, which aims to automatically generate or rewrite the instruction of the task. The guideline in this paper can also be viewed as part of the task instruction.**
>
> A: We tend to think of Guideline Learning (GL) as an innovative approach (as reviewer NFNq said) in the realm of instructional learning/induction. Our GL framework is designed to tackle complex tasks, learning guidelines iteratively from many error cases. In contrast, previous work, particularly the reference you provided, focuses on learning simple and clear instructions, prompting the LLM to predict the instruction contextually in a single pass. Specifically, GL differs from two main branches of previous work on instruction learning:
>  1. **Instruction induction via ICL.** Honovich et al. [2] predict the task instruction by prompting the instruction-tuned LLMs. They conduct explorative experiments, focusing on tasks that have "clear and simple instructions". Our GL framework focuses on more complex instructions with a highlight on IE tasks: extraction of complex concepts (event/relation). We propose the "guideline" as a bridge to learn and utilize more specific instructions from error cases automatically, which can be viewed as an in-depth extension of previous work.
>
> 2. **Instruction learning for meta-training.** Ye et al. [3] propose to utilize instruction learning to better finetune LLMs and boost the zero-shot performance. Our GL framework aims at boosting the model performance under the tuning-free setting, which is orthogonal to their work.
>
> To the best of our knowledge, we are the first to apply instruction learning (specifically guideline learning) to alleviate the "underspecified task description" problem for information extraction. (Please note that we have revised the phrase "conceptual bias" to "underspecified task description", following the suggestion from reviewer B35m.)
>
> **Q: Few datasets have rule-based guidelines, which limits the application of the framework. & Question E: Can the authors think of a way to apply this framework to datasets that do not have readily available guidelines?**
>
> A: It seems there may be a misunderstanding about our framework. It's important to note that our framework **does NOT** require the readily available guidelines of datasets. Our framework learns guidelines automatically from error cases. GL can be easily transferred to other datasets just with simple modifications on prompts. We've provided our prompts in the appendix for replication.
>
> **Q: Because the rule scoring and rule update rely on the true label y, instance selection can only be done over the instances that have labels. The authors should note that this is not true few-shot learning.**
>
> A: There is probably a misunderstanding. Our active instance selection **does NOT** require instance labels. We select instances according to the model's confidence in predicting the instance. The model's confidence is computed by measuring the consistency of multiple sampled responses of the model, where the true label is not necessary (detailed in Sec 3.4).
>
> **Q: Have the authors thought about the faithfulness of the rules returned by the Reasoner? Are these rules truly relevant to the instance and are they actually used by the LLM?**
>
> A: We conduct an additional experiment. Specifically, we manually evaluate the following aspects of responses from Reasoner: 1. **relevant**: whether the rules referred to by the model are truly relevant to the instance; 2. **well-referred**: whether the model genuinely follows the rules, i.e. the response is consistent with the rules it refers to. We analyze 50 responses generated by the gpt-3.5-turbo and gpt-4 agents. The results (acc) are shown below:
>
> | model | relevant | well-referred |
> | --- | --- | --- |
> | gpt-3.5-turbo | 0.84 | 0.88 |
> | gpt-4 | 0.94 | 0.98 |
>
> We find that gpt-3.5-turbo is capable of figuring out and following relevant rules, while gpt-4, known as the most powerful LLM, makes fewer mistakes. We utilize gpt-3.5-turbo as the backbone LLM in our experiments, which is sufficient to verify our framework. Moreover, our framework may potentially gain additional advantages from the increasing rule-following capabilities of these backbone LLMs.
>
> **Q: Can you provide some intuition on why the generalizer is needed or show some results that ablate this component?**
>
> A: The intuition of the generalizer is in two folds. First, the guideline should have some generalizability to cover/handle similar cases. Secondly, in practice, the generalizer is helpful to the guideline retrieval task based on the input case. If the guidelines are composed of corrected error cases, the retrieval would be case-to-case, which is very sensitive. For example in the following quote block, the input case is more similar to G1 literally. However, G2 is more relevant as they both describe an active underweight event. If we generate their general form by abstracting common properties (company name, number of shares, date), it will be more similar to G2.
>
> > **Input case**: Xinguang Investment actively reduced its shareholdings of this company by 300,000 shares.
> >
> > **G1**: Xinguang Investment passively reduced its shareholding in the company by 300,000 shares. This does not trigger an EU event.
> >
> > **G2**: Jinying Technology actively reduced its shareholdings of this company by 200,000 shares today. This triggers an EU event.
> >
> > **The General form of input case**: One company actively reduced its shareholdings of another company.
> >
> > **The General form of G1**: One company passively reduced its shareholdings of another company. This does not trigger an EU event.
> >
> > **The General form of G2**: One company actively reduced its shareholdings of another company. This triggers an EU event.
> >
> In our experiments, we implement the generalizer by few-shot prompting an LLM agent. Though the generalizer is critical for the GL framework, we don't put the implementation details into section 3 as there may be other underlying implementations, for example, finetuning a more effective generalizer, and we want to highlight our contribution on the guideline learning framework.
>
> **Q: In Table 2, it seems that guideline learning is not helpful for multi-event documents (75.0 F1 for ICL vs 74.4 F1 for GL), can the authors explain this behavior?**
>
> A: Though EE-GL-r is slightly below EE-ICL (74.4 vs 75.0), EE-GL-a surpasses EE-GL-r by a relatively large margin (76.2 vs 74.4). We suppose the reason is that some learned harmful guidelines in EE-GL-r slightly hurt the performance.
>
> **Q: Since some of the experiment settings (EE-GL-r, RE-GL-r) rely on random sampling in-context examples, the variance should be reported for results.**
>
> A: Thanks for your suggestion. Since manually annotating more examples requires some time, we will report the variance after we finish it.
>
> **If you have any further questions or concerns, please do not hesitate to inform us.**
>
> **Best regards.**
>
> [1] Memory-assisted prompt editing to improve GPT-3 after deployment. EMNLP 2022.
>
> [2] Instruction Induction: From Few Examples to Natural Language Task Descriptions. ACL 2023.
>
> [3] Guess the Instruction! Flipped Learning Makes Language Models Stronger Zero-Shot Learners. ICLR 2023.

---

### Meta-Review · Area_Chair_Pnsx · 2023-09-21

**Recommendation:** 4

**Metareview:**

The paper proposes a new model for guideline learning/refinement to enhance in-context learning. The idea is to provide a few training examples and a set of rules (guidelines). If the model makes an incorrect prediction, it updates the rules. The updated rules are then used for model inference.

The paper makes significant contributions to an active area of research and would be of considerable interest to the NLP community. The idea pursued in the paper is intuitive and clear, and it is convincingly demonstrated.

The paper lacks a mandatory "Limitations" section, which violates the EMNLP policy and could result in desk rejection. However, the authors have included text for Limitations section as part of the discussion and will include in the next revision. Moreover, as part of the discussion, the authors conducted additional experiments.The paper is well-written and easy to follow, with only a few typos and missing references.

---

### Decision · Program_Chairs · 2023-10-07

**Decision:**

Accept-Main

**Comment:**

The paper proposes a new model for guideline learning/refinement to enhance in-context learning. The idea is to provide a few training examples and a set of rules (guidelines). If the model makes an incorrect prediction, it updates the rules. The updated rules are then used for model inference.

The paper makes significant contributions to an active area of research and would be of considerable interest to the NLP community. The idea pursued in the paper is intuitive and clear, and it is convincingly demonstrated.

The paper lacks a mandatory "Limitations" section, which violates the EMNLP policy and could result in desk rejection. However, the authors have included text for Limitations section as part of the discussion and will include in the next revision. Moreover, as part of the discussion, the authors conducted additional experiments.The paper is well-written and easy to follow, with only a few typos and missing references.